# Association of lipid-lowering drugs with COVID-19 outcomes from a Mendelian randomization study

**Wuqing Huang[1†], Jun Xiao[2,3†], Jianguang Ji[4]\*, Liangwan Chen[2,3]\***

[1]Department of Epidemiology and Health Statistics,School of Public Health, Fujian Medical University, Fuzhou, China; [2]Department of Cardiovascular Surgery, Fujian Medical University Union Hospital, Fuzhou, China; [3]Key Laboratory of Cardio-Thoracic Surgery, Fujian Medical University, Fujian Province, Fuzhou, China; [4]Center for Primary Health Care Research, Department of Clinical Sciences, Lund University, Malmö, Sweden

## Abstract

**Background:** Lipid metabolism plays an important role in viral infections. We aimed to assess the causal effect of lipid-lowering drugs (HMGCR inhibitiors, PCSK9 inhibitiors, and NPC1L1 inhibitior) on COVID-19 outcomes using two-sample Mendelian randomization (MR) study.

**Methods:** We used two kinds of genetic instruments to proxy the exposure of lipid-lowering drugs, including expression quantitative trait loci of drugs target genes, and genetic variants within or nearby drugs target genes associated with low-density lipoprotein (LDL cholesterol from genome-wide association study). Summary-data-based MR (SMR) and inverse-variance-weighted MR (IVW-MR) were used to calculate the effect estimates.

**Results:** SMR analysis found that a higher expression of HMGCR was associated with a higher risk of COVID-19 hospitalization (odds ratio [OR] = 1.38, 95% confidence interval [CI] = 1.06–1.81). Similarly, IVW-MR analysis observed a positive association between HMGCR-mediated LDL cholesterol and COVID-19 hospitalization (OR = 1.32, 95% CI = 1.00–1.74). No consistent evidence from both analyses was found for other associations.

**Conclusions:** This two-sample MR study suggested a potential causal relationship between HMGCR inhibition and the reduced risk of COVID-19 hospitalization.

**Funding:** Start-up Fund for high-level talents of Fujian Medical University.

**\*For correspondence:**
jianguang.ji@med.lu.se (JJ);
chenliangwan@fjmu.edu.cn (LC)

[†]These authors contributed equally to this work

**Competing interest:** The authors declare that no competing interests exist.

## Editor's evaluation

There are mixed results from studies of COVID-19 outcomes in patients treated with statins and there are multiple confounders. The authors use two Mendelian randomization methods to explore the association between HMGCoA reductase inhibitors (statins) and other lipid lowering drugs and outcomes and find that increased expression of HMGCoA reductase and HMGCoA reductase mediated LDL cholesterol increase hospitalization risk. This makes it possible but does not prove that statins could improve outcomes which will be of broad interest.

## Introduction

The COVID-19 pandemic has caused millions of infections and deaths, which is caused by a novel coronavirus, severe acute respiratory syndrome coronavirus 2 (SARS-CoV-2). Lacking drugs specifically

**eLife digest** The virus SARS-CoV-2 has caused millions of infections and deaths during the COVID-19 pandemic, but as of December 2021, no new drugs targeted to SARS-CoV-2 specifically exist. Thus, it is important to identify existing drugs that can reduce the infection and mortality of this virus, since repurposing old drugs is faster and cheaper than developing new ones. Fats, such as cholesterol, can play an important role in viral infections, meaning that drugs intended to lower the levels of fats in the blood could have a protective effect against SARS-CoV-2.

To test this hypothesis, Huang, Xiao, et al. carried out a Mendelian randomization study to investigate if there is a link between drugs that lower fats and outcomes of SARS-CoV-2 infection, including susceptibility, hospitalization, and severe disease. This approach consists on grouping people according to their version of a particular gene, which minimizes the effect of variables that can cause spurious associations, something known as confounding bias. Thus, Mendelian randomization studies allow scientists to disentangle cause and effect.

Using this method, Huang, Xiao, et al. found an association between statins (a type of drug that decreases the levels of bad cholesterol) and a reduced risk of being hospitalized after being infected withSARS- CoV-2.

These findings suggest that statins could benefit patients infected with SARS-CoV- 2, and indicate that they should be prioritized in future clinical trials for treating COVID-19.

targeted to SARS-CoV-2 infection has led to a great interest to identify drugs that can be repurposed to reduce the infection and mortality of the disease.

Available studies have suggested an important role of lipid metabolism in viral infections, including in the pathogenesis of SARS-CoV-2 infection (*Proto et al., 2021*). The plausible mechanisms include the involvement of host lipids in the virus life cycle, the influence of cholesterol on the immune cell functions, interfering with the mevalonate pathway, and so on *Proto et al., 2021*. Such evidence indicates the potential protective effect of lipid-lowering drugs against COVID-19. HMG-CoA reductase (HMGCR) inhibitors, known as statins, are the most commonly used class of lipid-lowering drugs, which have a couple of predominant merits, such as the well-proven safety, low cost, and pleiotropic effects. Proprotein convertase subtilisin/kexin type 9 (PCSK9) and Niemann–Pick C1-Like 1 (NPC1L1) are proteins playing a crucial role for the circulating level of low-density lipoprotein cholesterol (LDL-C) (*Sabatine, 2019*; *Williams et al., 2020*). Both PCSK9 inhibitors (i.e., evolocumab and alirocumab) and NPC1L1 inhibitors (i.e., ezetimibe) are FDA-approved lipid-lowering agents (*Sabatine, 2019*; *Williams et al., 2020*). A number of observational studies have investigated the association between lipid-lowering drugs and COVID-19 outcomes, but generated mixed results (*Butt et al., 2020*; *Hariyanto and Kurniawan, 2020*; *Kow and Hasan, 2020*; *Zhang et al., 2020*; *Gupta et al., 2021*). What's more, confounding bias and reverse causation cannot be avoided in most of these studies.

Mendelian randomization (MR) study uses genetic variants as an instrument to perform causal inference between an exposure and an outcome, which could indicate whether an observational association is consistent with a causal effect (*Davies et al., 2018*). Confounding bias can be minimized in MR study because genetic variants are randomly assigned to the individual at birth. Similarly, reverse causation can be avoided because genetic variants are assigned prior to the development of disease.

Therefore, we performed two-sample MR analysis in this study to test the association of lipid-lowering drugs (HMGCR inhibitiors, PCSK9 inhibitiors, and NPC1L1 inhibitior) with COVID-19 outcomes (susceptibility, hospitalization and very severe disease).

## Materials and methods
### Study design
This two-sample MR study is based on publicly available summary-level data from genome-wide association studies (GWASs) and expression quantitative trait loci (eQTLs) studies (*Supplementary file 1*—Table 1). All these studies had been approved by the relevant institutional review boards and participants had provided informed consents.

**Table 1.** Information of genetic instruments.

Abbreviations and acronyms: eQTLs, expression quantitative trait loci; GWAS, genome-wide association study; HEIDI, heterogeneity in dependent instruments; HMGCR, HMG-CoA reductase; IVW-MR, inverse-variance-weighted Mendelian randomization; LDL, low-density lipoprotein; MR-PRESSO, Mendelian Randomization Pleiotropy RESidual Sum and Outlier; MAF, minor allele frequency; NPC1L1, Niemann–Pick C1-Like 1; PCSK9, proprotein convertase subtilisin/kexin type 9; SNP, single-nucleotide polymorphism; SMR, summary-data-based Mendelian randomization.

| Exposure | Genetic instruments | |
| --- | --- | --- |
| | Genetic variants associated with mRNA expression levels (eQTLs) | Genetic variants associated with LDL cholesterol level |
| HMGCR inhibitors | Nine hundred and twenty-one common cis-eQTLs (MAF >1%) in blood for HMGCR gene (p < 5.0 × 10⁻⁸), top SNP: rs6453133 | Seven common SNPs (MAF >1%) in low linkage disequilibrium ($r^2 < 0.30$), associated with LDL cholesterol (p < 5.0 × 10⁻⁸), located within ±100 kb windows from HMGCR region |
| PCSK9 inhibitors | Twenty-four common cis-eQTLs (MAF >1%) in blood for PCSK9 gene (p < 5.0 × 10⁻⁸), top SNP: rs472495 | Twelve common SNPs (MAF >1%) in low linkage disequilibrium ($r^2 < 0.30$), associated with LDL cholesterol (p < 5.0 × 10⁻⁸), located within ±100 kb windows from PCSK9 region |
| NPC1L1 inhibitors | Eleven common cis-eQTLs (MAF >1%) in adipose subcutaneous tissue for NPC1L1 gene (p < 5.0 × 10⁻⁸), top SNP: rs41279633 | Three common SNPs (MAF >1%) in low linkage disequilibrium ($r^2 < 0.30$), associated with LDL cholesterol (p < 5.0 × 10⁻⁸), located within ±100 kb windows from NPC1L1 region |

| Statistical analyses | | |
| --- | --- | --- |
| Primary analysis | Summary-data-based Mendelian randomization | Inverse-variance-weighted Mendelian randomization |
| Sensitivity analyses | *F*-Statistic<br>Positive control analysis (LDL cholesterol used as outcome)<br>Linkage disequilibrium test: HEIDI test<br>Horizontal pleiotropy test: SMR association between expression of adjacent genes and outcome | *F*-Statistic<br>Positive control analysis (coronary heart disease used as outcome)<br>Heterogeneity test: Cochran *Q* test<br>Horizontal pleiotropy test: MR-Egger regression, MR-PRESSO test |

## Selection of genetic instruments

Three classes of FDA-approved lipid-lowering drugs were included as exposures in this study: HMGCR inhibitors, PCSK9 inhibitors, and NPC1L1 inhibitor.

As shown in *Table 1*, we used available eQTLs for drugs target genes (i.e., HMGCR, PCSK9, and NPC1L1) as the proxy of exposure to each lipid-lowering drug. The eQTLs summary-level data were obtained from eQTLGen Consortium (https://www.eqtlgen.org/) or GTEx Consortium V8 (https://gtexportal.org/), the details of which are presented in *Supplementary file 1*—Table 1. We identified common (minor allele frequency [MAF] >1%) eQTLs single-nucleotide polymorphisms (SNPs) significantly (p < 5.0 × 10⁻⁸) associated with the expression of HMGCR or PCSK9 in blood, and the expression of NPC1L1 in adipose subcutaneous tissue as there are no eQTLs in blood or other tissues available at a significance level for NPC1L1. Only cis-eQTLs were included to generate genetic instruments in this study, which were defined as eQTLs within 1 Mb on either side of the encoded gene.

Secondly, to validate the observed association using the eQTLs as an instrument, we additionally proposed an instrument by selecting SNPs within 100 kb windows from target gene of each drug that was associated with LDL cholesterol level at a genome-wide significance level (p < 5.0 × 10⁻⁸) to proxy the exposure of lipid-lowering drugs. A GWAS summary data of LDL cholesterol levels from the Global Lipids Genetics Consortium (GLGC) with a sample size of 173,082 were used to identify these SNPs, where only common SNPs (MAF >1%) were included (*Willer et al., 2013*; *Supplementary file 1*—Table 1). Seven SNPs within 100 kb windows from HMGCR gene were selected for proxying HMGCR

inhibitors, 12 SNPs from PCSK9 gene identified for PCSK9 inhibitors, and 3 SNPs from NPC1L1 gene selected for NPC1L1 inhibitor. To maximize the strength of the instrument for each drug, SNPs used as instruments were allowed to be in low weak linkage disequilibrium ($r^2 < 0.30$) with each other.

## Outcome sources

GWAS summary-level data for COVID-19 outcomes were obtained from the COVID-19 Host Genetics Initiative V4 with a sample size of 1,299,010 for COVID-19 susceptibility, 908,494 for COVID-19 hospitalization, and 626,151 for COVID-19 severe disease, respectively (https://www.covid19hg.org/; *COVID-19 Host Genetics Initiative, 2020*; *Supplementary file 1*—Table 1). The study population was restricted to individuals with European ancestry, including meta-analyses of GWASs containing up to 22 cohorts from 11 countries. GWAS from these cohorts used a model adjusted for age, sex, age × age, age × sex, genetic principal components, and study-specific covariates. A COVID-19 case was confirmed by lab or self-reported infections, or electronic health records of infections. The suscepti-bility outcome was measured by comparing COVID-19 cases and controls who did not have a history of COVID-19. The hospitalized outcome was measured by comparing COVID-19 hospitalized cases and controls who were never admitted to the hospital due to COVID-19, including individuals without COVID-19. The severe disease outcome was measured by comparing COVID-19 cases who died or required respiratory support and controls without severe COVID-19, including individuals without COVID-19. We included individuals without COVID-19 as controls for all outcomes to decrease collider bias and allow for population-level comparisons (*Griffith et al., 2020*; *Butler-Laporte et al., 2021*).

## Statistical analyses

### Primary MR analysis

Summary-data-based MR (SMR) method was applied to generate effect estimates when using eQTLs as an instrument, which investigates the association between the expression level of a gene and outcome of interest using summary-level data from GWAS and eQTL studies (*Zhu et al., 2016*). Allele harmonization and analysis were performed using SMR software, version 1.03 (https://cnsgenomics.com/software/smr/#Overview). Inverse-variance-weighted MR (IVW-MR) method was used to combine effect estimates when using genetic variants associated with LDL cholesterol level as an instrument. Allele harmonization and analysis were conducted using the TwoSampleMR package in R software, version 4.1.0.

### Sensitivity analysis

The strength of SNPs used as the instrument was assessed using the *F*-statistic, and we included SNPs with an *F*-statistic of >10 to minimize weak instrument bias (*Burgess and Thompson, 2011*). Positive control analyses were performed for validation of both genetic instruments. Since lowering the level of LDL cholesterol is the well-proven effect of lipid-lowering drugs, we thus examined the associa-tion of exposures of interest with LDL cholesterol level as positive control study for the instrument from eQTLs. For the instrument from LDL cholesterol GWAS, we performed positive control study by examining the association of exposures of interest with coronary heart disease because coronary heart disease is the main indication of lipid-lowering drugs.

For SMR method, the heterogeneity in dependent instruments (HEIDI) test was used to test if the observed association between gene expression and outcome was due to a linkage scenario, which was performed in the SMR software (*Zhu et al., 2016*). The HEIDI test of p < 0.01 indicates that association is probably due to linkage (*Chauquet et al., 2021*). One SNP could be related to the expression of more than one genes, leading to the presence of horizontal pleiotropy. To assess the risk of horizontal pleiotropy, we identified other nearby genes (within a 1 Mb window), the expression of which was significantly associated with the genetic instrumental variant, and performed SMR analysis to examine if the expression of these genes was related to the COVID-19 outcomes.

For IVW-MR method, we tested the heterogeneity by using a Cochran *Q* test, where p < 0.05 indicates the evidence of heterogeneity (*Higgins et al., 2003*). MR-Egger regression and Mendelian Randomization Pleiotropy RESidual Sum and Outlier (MR-PRESSO) analysis were used to assess the potential horizontal pleiotropy of the SNPs used as instrument variants. In MR Egger regression, the intercept term is a useful indication of directional horizontal pleiotropy, where p < 0.05 indicates the evidence of horizontal pleiotropy (*Burgess and Thompson, 2017*). MR-PRESSO analysis can identify

horizontal pleiotropic outliers and provide adjusted estimates, where p < 0.05 for Global test indicates the presence of horizontal pleiotropic outliers (*Verbanck et al., 2018*). Besides, a multivariable MR study was further conducted to examine if the observed association was direct association. We first investigated the association of HMGCR-mediated LDL cholesterol with the common risk factors of COVID-19 hospitalization, including body mass index, diabetes, hypertension, and coronary heart disease. After that, a multivariable MR study was performed by adjusting for the factors which showed a significant association. All these analyses were implemented in R software, version 4.1.0.

To account for multiple testing, Bonferroni correction was used to adjust the thresholds of significance level, thus a strong evidence was suggested for p < 0.006 (three exposures and three outcomes) and a suggestive evidence of 0.006 ≤ p < 0.05.

## Results

### Genetic instruments selection and COVID-19 outcomes

A total of 921,24, and 11 cis-eQTLs were identified from eQTLGen or GTEx Consortium for drugs target gene HMGCR, PCSK9, and NPC1L1, respectively, and the most significant cis-eQTL SNP was selected as a genetic instrument for the target gene of each drug (*Table 1*, *Supplementary file 1*—Table 2). A total of 7, 12, and 3 SNPs within or nearby gene HMGCR, PCSK9, and NPC1L1 were selected from a GWAS summary data of LDL cholesterol levels in the Global Lipids Genetics Consortium, respectively (*Table 1*, *Supplementary file 1*—Table 3). *F*-Statistics for all instrument variants were over 30, suggesting that weak instrument bias can be minimized in our study (*Supplementary file 1*-Tables 2 and 3). Positive control study showed significant associations between exposure to each drug and LDL cholesterol when using eQTLs-proposed instruments (*Supplementary file 1*—Table 5), as well as between exposure to each drug and coronary heart disease when using LDL cholesterol GWAS-proposed instruments (*Supplementary file 1*—Table 6), further ensuring the efficacy of the selected genetic instruments.

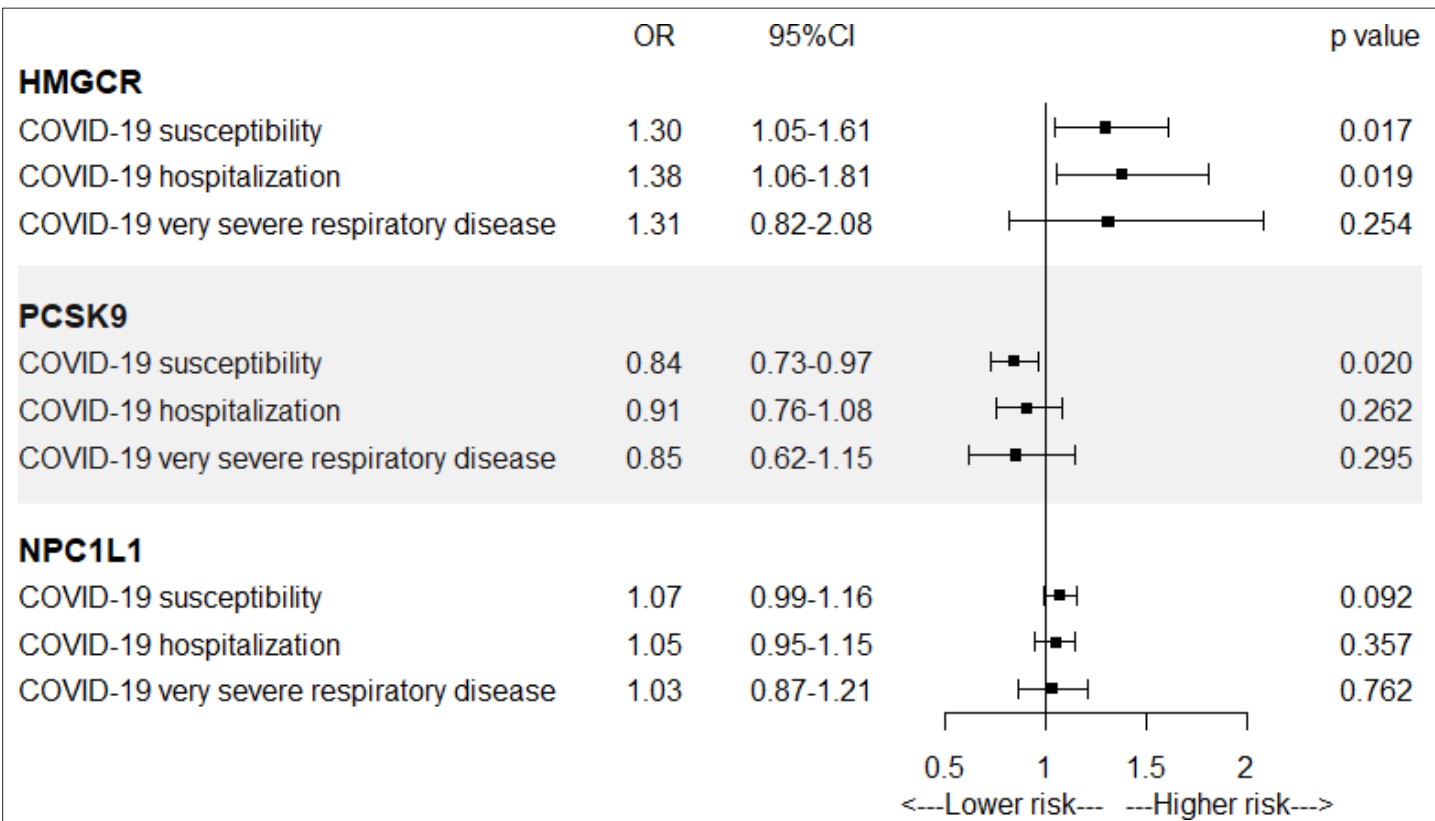

**Figure 1.** Summary-data-based Mendelian randomization (SMR) association between expression of gene HMGCR, PCSK9, or NPC1L1 and COVID-19 outcomes. SMR method was used to assess the association.

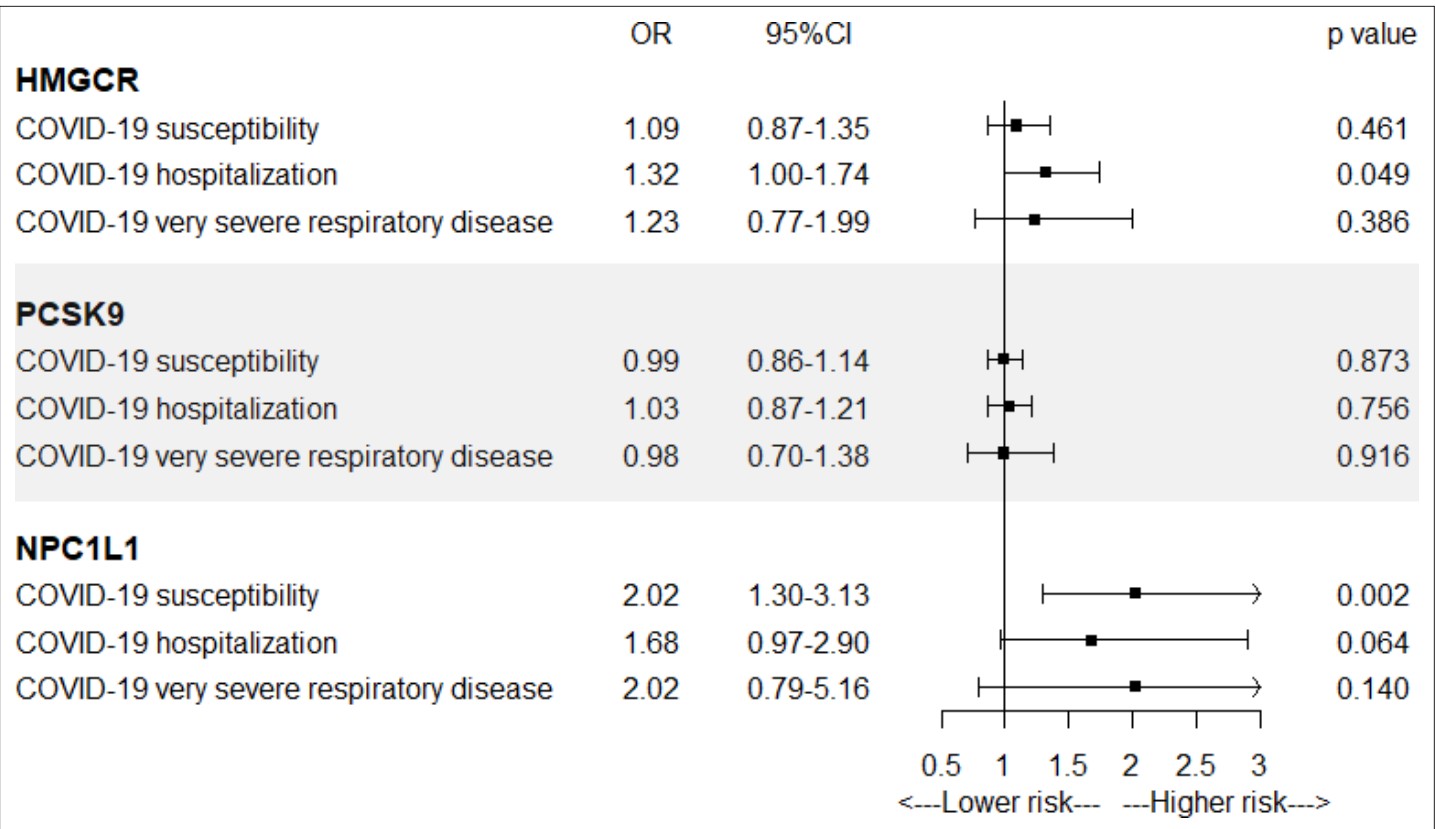

**Figure 2.** Inverse-variance-weighted Mendelian randomization (IVW-MR) association between low-density lipoprotein (LDL) cholesterol mediated by gene HMGCR, PCSK9, or NPC1L1 and COVID-19 outcomes. IVW-MR method was used to assess the association.

From COVID-19 GWASs, a total of 14,134 cases and 1,284,876 controls were used to explore the association with COVID-19 susceptibility, 6406 cases and 902,088 controls for COVID-19 hospitalization, and 3886 cases and 622,265 controls for COVID-19 severe disease (*Supplementary file 1*—Table 1).

### Primary analysis

In *Figure 1* and *Supplementary file 1*—Table 2, results from SMR analysis found a suggestive evidence for the association of the increased expression of HMGCR gene in blood (equivalent to a one standard deviation increase) with the higher risk of COVID-19 susceptibility (odds ratio [OR] = 1.30, 95% confidence interval [CI] = 1.05–1.61; p = 0.017) and COVID-19 hospitalization (OR = 1.38, 95% CI = 1.06–1.81; p = 0.019), indicating that HMGCR inhibitors might lower the risk of COVID-19 susceptibility and hospitalization. Suggestive evidence was observed regarding the negative association between PCSK9 expression and risk of COVID-19 susceptibility (OR = 0.84, 95% CI = 0.73–0.97; p = 0.02). No significant association was found between the expression of NPC1L1 and COVID-19 outcomes.

In *Figure 2* and *Supplementary file 1*—Table 4, IVW-MR analysis also found a suggestive evidence for the association between HMGCR-mediated LDL cholesterol (equivalent to a 1 mmol/l increase) and the risk of COVID-19 hospitalization (OR = 1.32, 95% CI = 1.00–1.74; p = 0.049), further supporting a possible protective effect of HMGCR inhibitors against COVID-19 hospitalization. Strong evidence was observed between NPC1L1-mediated LDL cholesterol and the risk of COVID-19 susceptibility (OR = 2.02, 95% CI = 1.30–3.13; p = 0.002). IVW-MR analysis did not provide any evidence for the association between PCSK9-mediated LDL cholesterol and COVID-19 outcomes.

### Sensitivity Analysis

For SMR analysis, HEIDI test suggested that all observed associations were not due to a linkage (p > 0.01), except for the association between HMGCR expression and COVID-19 susceptibility (p = 0.009) (*Supplementary file 1*—Table 2). We further examine if horizontal pleiotropy was present in

the association between HMGCR expression and COVID-19 outcomes by investigating if there was an association between the expression of nearby genes which are significantly associated with the top eQTL SNP (instrument variant) of HMGCR and COVID-19 outcomes. We identified six genes, including HMGCR, the expression of which were associated with the instrument variant (*Supplementary file 1*—Table 7). Only four genes have available eQTLs at a genome-wide significance level (p < $5.0 \times 10^{-8}$). Among these four genes, only HMGCR expression was significantly related to COVID-19 susceptibility and COVID-19 hospitalization, suggesting a small role of horizontal pleiotropy in the observed associations (*Supplementary file 1*—Table 8).

For IVW-MR analysis, Cochran Q test did not find evidence of heterogeneity for all reported results (all p > 0.05; *Supplementary file 1*—Table 4). Both the intercept term in MR-Egger regression and MR-PRESSO analysis suggested no significant overall horizontal pleiotropy (all p > 0.05; *Supplementary file 1*—Table 4). A multivariable MR study suggested that BMI and diabetes might play a role in the association between HMGCR-mediated LDL cholesterol level and COVID19 hospitalization (*Supplementary file 1*—Tables 9 and 10).

## Discussion

The present MR study provided a suggestive evidence regarding the positive association of the HMGCR expression and HMGCR-mediated LDL cholesterol level with the risk of COVID-19 hospitalization, both of which together indicated a potential protective effect of HMGCR inhibition against COVID-19 hospitalization (OR $_{instrument\ 1}$ = 0.72, 95% CI = 0.55–0.95; OR $_{instrument\ 2}$=0.76, 95% CI = 0.57–1.00). We found a suggestive evidence of the negative association between PCSK9 expression and COVID-19 susceptibility, but which was not supported when using LDL cholesterol GWAS as an instrument. A strong evidence was observed for the protective effect of NPC1L1-mediated lower level of LDL cholesterol on COVID-19 susceptibility, but there was no evidence for the association of NPC1L1 expression and COVID-19 outcomes.

Compared to developing a new drug, repurposing an old drug is much more economical and time-saving, in particular, the importance is further highlighted during a pandemic. COVID-19 pandemic has driven a number of studies of drug repurposing (*Fajgenbaum and Rader, 2020*; *Gaziano et al., 2021*). The role of lipid metabolism in viral infections has raised interest regarding the possibility of repurposing lipid-lowering drugs as anti-COVID-19 agents (*Proto et al., 2021*). As one of the most commonly prescribed drugs, statins have received the greatest attention for their pleiotropic effects, including lowering serum cholesterol, anti-inflammatory and immunomodulatory properties, and antithrombotic effect, all of which play a role in viral infections (*Fajgenbaum and Rader, 2020*; *Proto et al., 2021*; *Rubin, 2021*). Emerging observational studies have investigated if statins might benefit patients with COVID-19 (*Butt et al., 2020*; *Hariyanto and Kurniawan, 2020*; *Kow and Hasan, 2020*; *Zhang et al., 2020*; *Gupta et al., 2021*). A largest retrospective cohort study including 13,981 patients admitted to hospital due to COVID-19 suggested a significant reduction in 28-day all-cause mortality by 42% in the group with statins than patients without statins (*Zhang et al., 2020*). A meta-analysis with 8990 COVID-19 patients also found a 30% lower risk of fatal or severe disease (*Kow and Hasan, 2020*). However, a Danish nationwide cohort study with 4842 COVID-19 patients and a meta-analysis with 3449 COVID-19 patients did not find the association between statins use and improved COVID-19 outcomes (*Butt et al., 2020*; *Hariyanto and Kurniawan, 2020*). In addition, in these studies, considerable differences in clinical characteristics cannot be avoided between patients with and without statins, and causal inference is not allowed due to the retrospective nature of observational studies.

As a genetic epidemiological method, MR study could overcome the limitations of traditional observational studies. In this MR study, we used genetic variants related to HMGCR expression or HMGCR-mediated LDL cholesterol as instruments to proxy the exposure of statins. Both analyses found a suggestive evidence that HMGCR inhibition could reduce the risk of COVID-19 hospitalization, rather than COVID-19 susceptibility and very severe outcome. Although strong evidence is lacking, these results provided a causal evidence supporting the finding from the largest cohort study (*Zhang et al., 2020*), which calls for additional observational studies in different populations, mechanistic studies, and randomized controlled studies to examine its potential effect against COVID-19. Patients with COVID-19 who already take statins or start to take it for the indication of statins were recommended to continue to take it, which might be beneficial to both its original indication and COVID-19 (*Rubin, 2021*). And statins might be a prioritized drug in future clinical trials for treating COVID-19.

Besides, although no association was found between NPC1L1 expression in adipose subcutaneous and COVID-19 outcomes, there was a strong evidence of the association between NPC1L1-mediated LDL cholesterol and COVID-19 susceptibility. The effect of NPC1L1 inhibitor on COVID-19 susceptibility may be worth further studies as well.

## Study strengths

The main strength of our study is the use of genetic instruments to proxy drug exposure, which could minimize confounding bias and avoid reverse causation. Besides, we used two different kinds of genetic instruments to proxy the studied drug, which contributes to validating the effect estimates from each other. A number of sensitivity analyses have been performed to test the efficacy of genetic instruments and the assumptions of MR study.

## Study limitations

This study has several limitations. Firstly, there are no available eQTLs in blood for NPC1L1, so we were not able to explore the association between NPC1L1 expression in blood and COVID-19 outcomes. Besides, there are no available eQTLs in liver (the main tissue related to lipid metabolism) for these target genes, which might provide more convincing evidence of the observed association. The sample size of eQTL study for PCSK9 and NPC1L1 in GTEx is relatively smaller, which may affect the statistical power for the results of PCSK9 or NPC1L1 inhibition. Secondly, the effect of statins probably varies between subgroups, for example, it may be more effective in patients with chronic diseases (e.g., coronary heart disease). However, the use of summary-level data did not allow us to perform subgroup analyses, so further MR study with individual-level data is needed to provide more detailed information. Thirdly, the Bonferroni correction for multiple tests suggests that we cannot rule out the false-positive possibility for the finding of the protective effect of statins on COVID hospitalization. Fourthly, confounding bias and/or horizontal pleiotropy cannot be completely excluded although we have performed various sensitivity analyses to test the assumptions of MR study. Fifthly, both eQTLs and GWAS data used in this study were predominantly obtained from European population ancestry, thus these findings should be interpreted with caution when generalizing to other populations.

## Conclusions

In conclusion, this MR study suggested a causal relationship between HMGCR inhibition and the reduced risk of COVID-19 hospitalization. Clninical trials are called to examine if statins have the protective effect against COVID-19 and further researches are needed to explore the underlying mechanisms.

## Acknowledgements

We thank the patients and investigators who contributed to the eQTLGen Consortium, GTEx Consortium, COVID-19 Host Genetics Initiative, Global Lipids Genetics Consortium, and CARDIoGRAMplusC4D Consortium.

## Additional information

### Funding

| Funder | Grant reference number | Author |
| --- | --- | --- |
| Start-up Fund for high-level talents of Fujian Medical University | XRCZX2021026 | Wuqing Huang |

The funder had no role in study design, data collection, and interpretation, or the decision to submit the work for publication.

### Author contributions

Wuqing Huang, Conceptualization, Data curation, Formal analysis, Investigation, Methodology, Software, Validation, Visualization, Writing – original draft, Writing – review and editing; Jun Xiao,

Conceptualization, Formal analysis, Methodology, Software, Validation, Visualization, Writing – original draft, Writing – review and editing; Jianguang Ji, Conceptualization, Methodology, Project administration, Supervision, Validation, Visualization, Writing – review and editing; Liangwan Chen, Conceptualization, Data curation, Methodology, Project administration, Supervision, Validation, Visualization

**Author ORCIDs**
Wuqing Huang ⓘ http://orcid.org/0000-0002-7616-8622
Jun Xiao ⓘ http://orcid.org/0000-0002-5046-5493
Jianguang Ji ⓘ http://orcid.org/0000-0003-0324-9496
Liangwan Chen ⓘ http://orcid.org/0000-0002-4211-3842

**Ethics**
This two-sample MR study is based on publicly available summary-level data from genome-wide association studies (GWASs) and expression quantitative trait loci (eQTLs) studies. All these studies had been approved by the relevant institutional review boards and participants had provided informed consents.

**Decision letter and Author response**
Decision letter https://doi.org/10.7554/eLife.73873.sa1
Author response https://doi.org/10.7554/eLife.73873.sa2

---

## Additional files

**Supplementary files**
• Transparent reporting form
• Supplementary file 1. Source data and additional analyses.

**Data availability**
Individual-level data cannot be provided but the raw data of the eQTLGen Consortium, GTEx and COVID-19 Host Genetics Initiative can be acessed at https://www.eqtlgen.org/, https://gtexportal.org/, and https://www.covid19hg.org/, respectively. Summary-level GWAS or eQTL data and code used to produce main results have been uploaded to GitHub (https://github.com/WH57/lipid_covid19.git), (copy archived at https://archive.softwareheritage.org/swh:1:rev:3f6e94c8e0553595f6a011e701b01ec3d5380b72). All MR results and GWAS or eQTL associations of selected SNPs were provided in the Supplementary File 1 - Tables 2 to 4.

The following previously published datasets were used:

| Author(s) | Year | Dataset title | Dataset URL | Database and Identifier |
|---|---|---|---|---|
| Võsa Urmo | 2018 | The eQTLGen Consortium:Cis-eQTLs | https://www.eqtlgen.org/ | The eQTLGen Consortium, Cis-eQTLs |
| Consortium GTEx | 2017 | GTExV8 | https://gtexportal.org/ | GTEx Consortium, GTExV8 |
| COVID-19 Host Genetics Initiative. | 2020 | COVID-19 Host Genetics Initiative V4 | https://www.covid19hg.org/ | COVID-19 Host Genetics Initiative, V4 |
| Willer CJ | 2013 | Global Lipids Genetics Consortium | http://lipidgenetics.org/ | Global Lipids Genetics Consortium, GLGC |
| Nikpay M | 2015 | CARDIoGRAMplusC4D Consortium | http://www.cardiogramplusc4d.org/ | CARDIoGRAMplusC4D Consortium, NA |

---

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
