## [Editor Report]

There are mixed results from studies of COVID-19 outcomes in patients treated with statins and there are multiple confounders. The authors use two Mendelian randomization methods to explore the association between HMGCoA reductase inhibitors (statins) and other lipid lowering drugs and outcomes and find that increased expression of HMGCoA reductase and HMGCoA reductase mediated LDL cholesterol increase hospitalization risk. This makes it possible but does not prove that statins could improve outcomes which will be of broad interest.

---

## [Decision Letter]

**Decision letter after peer review:**

Thank you for submitting your article "Association of lipid-lowering drugs with COVID-19 outcomes from a Mendelian randomization study" for consideration by *eLife*. Your article has been reviewed by 2 peer reviewers, including Edward D Janus as Reviewing Editor and Reviewer #1, and the evaluation has been overseen by David Serwadda as the Senior Editor. The following individual involved in review of your submission has agreed to reveal their identity: Xia Jiang (Reviewer #2).

Essential revisions:

1) HMG-CoA reductase was introduced in a detailed manner in the first paragraph of introduction, while PCSK 9 inhibitor and NPC1L1 inhibitor were not mentioned until the last paragraph of introduction. The abbreviations were not spelt out. Please adjust the flow to make it more smoothly.

2) The authors demonstrate with SMR that a higher expression of HMGCR was associated with a higher risk of COVID-19 hospitalization. With IVW-MR they observed a positive association between HMGCR -mediated LDL cholesterol and COVID-19 hospitalization. For the other two drug classes no clear associations were shown.

This suggests but does not prove that HMGCR drugs ie statins might improve outcomes. For the other two classes its unclear if there would be sufficient users of these drugs in the data sets to provide enough power to adequately address the issue.

3) Overall this is a well conducted study providing important further insights but the final sentence of the abstract as well as the related Discussion section including the limitations should be more expressed more conservatively.

4) Choosing the right tissue is very important for eQTL analysis. While a wide number of tissues are available from GTEx consortium, the authors focused only on the blood or adipose subcutaneous tissues. Please clarify the rationale. How about cardiovascular tissues, liver and other organs? have the authors considered other relevant tissues available in GTEx?

5) As mentioned in the methods, "only cis-eQTLs were included to generate genetic instruments in this study, which were defined as eQTLs within 1 Mb on either side of the encoded gene" were included. Were these SNPs independent of each other? Have the authors considered linkage disequilibrium, and if so, what was the threshold? Including correlated instruments might bias the results (false positive).

6) Please clarify the ethnicity of population involved in the current study as well as the generalizability of results.

7) The method section lacked proper description on the GWAS data used for LDL and COVID19 outcome.

8) As mentioned by the authors, "we additionally proposed an instrument by selecting SNPs within 100 kb windows from target gene of each drug that were associated with LDL cholesterol level at a genome-wide significance level (p<5.0 × 10-8) to proxy the exposure of lipid-lowering drugs" – it means the authors intersect the eQTL with LDL-SNPs? What is the purpose of doing so, in other words, by choosing only part of the SNPs, would the MR assumption still hold?

9) The causal association between HMGCR and COVID19 susceptibility was suggestive, so was the causal association between PCSK9 and COVID19 susceptibility, while the estimates were in opposite direction (1.30 vs. 0.84) – how to interpret such results?

10) Have the authors considered confounding effect from, for example, obesity, to the identified putative causal relationship?

---

## [Author Response]

Essential revisions:1) HMG-CoA reductase was introduced in a detailed manner in the first paragraph of introduction, while PCSK 9 inhibitor and NPC1L1 inhibitor were not mentioned until the last paragraph of introduction. The abbreviations were not spelt out. Please adjust the flow to make it more smoothly.

Thank you for the suggestions, we have added descriptions of PCSK 9 inhibitor and NPC1L1 inhibitor following HMG-CoA reductase inhibitors in the first paragraph of introduction and full names at the place when first appeared in the manuscript. #Page4

2) The authors demonstrate with SMR that a higher expression of HMGCR was associated with a higher risk of COVID-19 hospitalization. With IVW-MR they observed a positive association between HMGCR -mediated LDL cholesterol and COVID-19 hospitalization. For the other two drug classes no clear associations were shown.This suggests but does not prove that HMGCR drugs ie statins might improve outcomes. For the other two classes its unclear if there would be sufficient users of these drugs in the data sets to provide enough power to adequately address the issue.

We agree that the current evidence is not enough to prove that HMGCR inhibitors might improve outcomes. We thus did some revisions on both abstract and discussion to interprete the results in a more conservative way #Page 3,11. We agree that the sample size of eQTL study for PCSK9 and NPC1L1 in GTEx is relatively smaller, which may affect the statistical power for the results of PCSK9 or NPC1L1 inhibition. We have added related discussions in the limitation section. #Page10

3) Overall this is a well conducted study providing important further insights but the final sentence of the abstract as well as the related Discussion section including the limitations should be more expressed more conservatively.

It is an important point. As our answer to the previous question, we have revised the abstract and related Discussion section accordingly.

4) Choosing the right tissue is very important for eQTL analysis. While a wide number of tissues are available from GTEx consortium, the authors focused only on the blood or adipose subcutaneous tissues. Please clarify the rationale. How about cardiovascular tissues, liver and other organs? have the authors considered other relevant tissues available in GTEx?

We agree that choosing the right tissue is important for eQTL analysis. Liver is the main tissue related to lipid metabolism, unfortunately there are no available eQTLs in liver for these target genes. Although a wide number of tissues are available from GTEx consortium, the eQTLs at a genome-wide significance level (p<5.0 × 10-8) are limited. For example, for HMGCR, there are only eQTLs identified in blood from eQTLGen and eQTLs identified in muscle skeletal tissue from GTEx. But sharing eQTLs across tissues is very common, particularly for top SNPs. For example, previous study found that genetic effects at the top cis-eQTLs were highly correlated between independent brain and blood samples ^1^. And eQTLs data from blood is most comprehensive due to the easier access to blood sample as compared to other specific tissues. We thus first focused on eQTLs from blood, which we believe could provide some clues. When the blood eQTLs data is unavailable, we then focused on adipose tissue, because it is another tissue related to lipid metabolism.

5) As mentioned in the methods, "only cis-eQTLs were included to generate genetic instruments in this study, which were defined as eQTLs within 1 Mb on either side of the encoded gene" were included. Were these SNPs independent of each other? Have the authors considered linkage disequilibrium, and if so, what was the threshold? Including correlated instruments might bias the results (false positive).

As we mentioned in the method section, HEIDI test was used to test if the observed association between gene expression and the outcome was due to a linkage disequilibrium. The HEIDI test of P<0.01 indicates that association is probably due to linkage. Briefly, if gene expression and a trait share the same causal variant, the b_xy_ values calculated for any SNPs in LD (using the default value of r2 > 0.05 and also r2 < 0.9 to avoid issues of collinearity) with the causal variant should be identical. Therefore, testing against this null hypothesis of a single causal variant is equivalent to testing for heterogeneity in the b_xy_ values estimated for the SNPs in the cis-eQTL region. The detailed information of the results from HEIDI test is presented in Supplementary file 1-Table 2.

6) Please clarify the ethnicity of population involved in the current study as well as the generalizability of results.

Thank you for the suggestion. The discussion about the ethnicity of population and the generalizability of results has been added in the limitation section. #Page11

7) The method section lacked proper description on the GWAS data used for LDL and COVID19 outcome.

We have added brief descriptions on the GWAS data and presented the detailed information in Supplementary file 1-Table 1. #Page12-13

8) As mentioned by the authors, "we additionally proposed an instrument by selecting SNPs within 100 kb windows from target gene of each drug that were associated with LDL cholesterol level at a genome-wide significance level (p<5.0 × 10-8) to proxy the exposure of lipid-lowering drugs" – it means the authors intersect the eQTL with LDL-SNPs? What is the purpose of doing so, in other words, by choosing only part of the SNPs, would the MR assumption still hold?

We apologize for the misunderstanding, we did not mean to intersect the eQTL with LDL-SNP. One genetic instrument in our analysis is SNPs located in the target gene of each drug that was associated with LDL cholesterol, because LDL-C is the most important cholesterol in disease development and it is well-proved that these lipid-lowering drugs are effective in lowering the plasm level of LDL-C. This approach has been applied in several previous studies ^2, 3^. The other instrument is SNPs associated with the expression of target gene of each drug (i.e., eQTLs) by using SMR method, which is an analysis to test if the effect size of SNP on the phenotype is mediated by gene expression. This approach has been also applied in previous studies ^4, 5^.

9) The causal association between HMGCR and COVID19 susceptibility was suggestive, so was the causal association between PCSK9 and COVID19 susceptibility, while the estimates were in opposite direction (1.30 vs. 0.84) – how to interpret such results?

We found a negative association between PCSK9 and COVID19 susceptibility in the SMR study by using SNPs associated expression of target genes as instrument. However, we know that PCSK9 needs to bind to the LDL receptor to adjust LDL cholesterol level, thus its functions might be affected by the expression of the LDL receptor. Additionally, no association was observed when using SNPs as the instrument of LDL cholesterol, suggesting lowering LDL cholesterol by targeting PCSK9 might not affect the outcomes of COVID-19. As for HMGCR, we found the associations using SNPs as the instruments for both the expression and LDL cholesterol with COVID-19 hospitalization. As HMGCR is the rate-limiting enzyme, it can directly affect the synthesizing of cholesterol, thus we conclude that lowering LDL cholesterol by targeting HMGCR might affect COVID-19 hospitalization.

10) Have the authors considered confounding effect from, for example, obesity, to the identified putative causal relationship?

We agree that risk factors of COVID-19 may have an impact on identifying a putative causal relationship, such as obesity, chronic diseases (eg., diabetes, hypertention, coronary heart disease). As suggested in previous MR study ^6^, the impact of proposed risk factor may be due to pleiotropic or confounding (Author response image 1). We thus added multivariable MR analyses for the observed association in the IVW-MR study by including covariates that were associated with HMGCR-mediated LDLC, (Supplementary file 1-Table 9,10) #Page 15. Such analyses could not distinguish between the pleiotropic and confounding, but it can provide an estimate of the direct association between HMGCR-mediated LDLC and COVID19 hospitalization.

**Author response image 1. sa2fig1:** DAG showing the hypothesized relationships.

**References**

1. Qi T, Wu Y, Zeng J, Zhang F, Xue A, Jiang L, Zhu Z, Kemper K, Yengo L, Zheng Z, e QC, Marioni RE, Montgomery GW, Deary IJ, Wray NR, Visscher PM, McRae AF and Yang J. Identifying gene targets for brain-related traits using transcriptomic and methylomic data from blood. *Nat Commun*. 2018;9:2282.

2. Yarmolinsky J, Bull CJ, Vincent EE, Robinson J, Walther A, Smith GD, Lewis SJ, Relton CL and Martin RM. Association Between Genetically Proxied Inhibition of HMG-CoA Reductase and Epithelial Ovarian Cancer. *Jama*. 2020;323:646-655.

3. Ference BA, Ray KK, Catapano AL, Ference TB, Burgess S, Neff DR, Oliver-Williams C, Wood AM, Butterworth AS, Di Angelantonio E, Danesh J, Kastelein JJP and Nicholls SJ. Mendelian Randomization Study of ACLY and Cardiovascular Disease. *The New England journal of medicine*. 2019;380:1033-1042.

4. Chauquet S, Zhu Z, O'Donovan MC, Walters JTR, Wray NR and Shah S. Association of Antihypertensive Drug Target Genes With Psychiatric Disorders: A Mendelian Randomization Study. *JAMA Psychiatry*. 2021;78:623-631.

5. Gaziano L, Giambartolomei C, Pereira AC, Gaulton A, Posner DC, Swanson SA, Ho YL, Iyengar SK, Kosik NM, Vujkovic M, Gagnon DR, Bento AP, Barrio-Hernandez I, Ronnblom L, Hagberg N, Lundtoft C, Langenberg C, Pietzner M, Valentine D, Gustincich S, Tartaglia GG, Allara E, Surendran P, Burgess S, Zhao JH, Peters JE, Prins BP, Angelantonio ED, Devineni P, Shi Y, Lynch KE, DuVall SL, Garcon H, Thomann LO, Zhou JJ, Gorman BR, Huffman JE, O'Donnell CJ, Tsao PS, Beckham JC, Pyarajan S, Muralidhar S, Huang GD, Ramoni R, Beltrao P, Danesh J, Hung AM, Chang KM, Sun YV, Joseph J, Leach AR, Edwards TL, Cho K, Gaziano JM, Butterworth AS, Casas JP and Initiative VAMVPC-S. Actionable druggable genome-wide Mendelian randomization identifies repurposing opportunities for COVID-19. *Nat Med*. 2021;27:668-676.

6. Kar SP, Brenner H, Giles GG, Huo D, Milne RL, Rennert G, Simard J, Zheng W, Burgess S and Pharoah PDP. Body mass index and the association between low-density lipoprotein cholesterol as predicted by HMGCR genetic variants and breast cancer risk. *Int J Epidemiol*. 2019;48:1727-1730.